# Genetic Background and Antibiotic Resistance Profiles of *K. pneumoniae* NDM-1 Strains Isolated from UTI, ABU, and the GI Tract, from One Hospital in Poland, in Relation to Strains Nationally and Worldwide

**DOI:** 10.3390/genes12081285

**Published:** 2021-08-22

**Authors:** Magdalena Wysocka, Roxana Zamudio, Marco R. Oggioni, Justyna Gołębiewska, Marek Bronk, Beata Krawczyk

**Affiliations:** 1Department of Molecular Biotechnology and Microbiology, Faculty of Chemistry, Gdańsk University of Technology, ul. Narutowicza 11/12, 80-233 Gdańsk, Poland; magwojta@student.pg.edu.pl; 2Department of Genetics and Genome Biology, University of Leicester, University Road, Leicester LE1 7RH, UK; roxanzamudio@gmail.com (R.Z.); mro5@leicester.ac.uk (M.R.O.); 3Department of Nephrology, Transplantology and Internal Medicine, Medical University of Gdańsk, ul. Dębinki 7, 80-952 Gdańsk, Poland; jgolebiewska@gumed.edu.pl; 4Laboratory of Clinical Microbiology, University Centre for Laboratory Diagnostics, Medical University of Gdańsk Clinical Centre, ul. Dębinki 7, 80-952 Gdańsk, Poland; msb@gumed.edu.pl

**Keywords:** new delhi metallo-β-lactamase, *Klebsiella pneumoniae*, UTI (urinary tract infections), WGS, epidemiology, evolution, ST11, antimicrobial resistance

## Abstract

In recent years, there has been an observed increase in infections caused by carbapenem-resistant *Klebsiella pneumonia* (Kp) strains. The aim of this study was the phenotypic and genotypic analysis of eight *K. pneumoniae* NDM (Kp NDM) isolates, recovered in Poland during the years 2016 and 2018 from seven patients with urinary tract infections (UTIs), asymptomatic bacteriuria (ABU), or colonization of the gut. PCR melting profile genotyping indicated a close relationship between the strains derived from 2018, which were not related to the strain isolated in 2016. WGS results were analyzed in relation to international Kp isolates. Clonal and phylogenetic analyses were performed based on multilocus sequence typing (MLST) and single nucleotide polymorphisms (SNPs) of the core genome. The metallo-β-lactamase was assigned to the NDM-1 type and the sequence was identified as ST11. Eleven antimicrobial resistance genes were detected, mostly from plasmid contigs. Unprecedented profiles of plasmid replicons were described with the IncFII/pKPX-1 dominant replicon. In terms of the KL24 and O2v1 capsular antigen profiles, these isolates corresponded to Greek strains. Strains isolated from UTI, ABU, and colonization GI tract patients were not carrying environment-specific virulence genes. Based on the assessment of strain relationships at the genome level and their direction of evolution, the international character of the sublines was demonstrated, with a documented epidemic potential in Poland and Greece. In conclusion, some groups of patients, e.g., renal transplant recipients or those with complicated UTIs, who are frequently hospitalized and undergoing antibiotic therapy, should be monitored not only for the risk of UTI, but also for colonization by Kp NDM strains.

## 1. Introduction

In recent years, there has been a substantial increase in the number of reported cases and outbreaks caused by *K. pneumoniae* (Kp) strains, both extended spectrum β-lactamase (ESBL) strains and Kp carbapenemase (KPC)- and metallo-β-lactamase (MBL)-producing strains [1,2]. The persistence of these carbapenemase-producing enterobacteriaceae (CPE) in the hospital environment and their ability to spread between patients make Kp strains a serious problem in European hospitals and around the world [3]. European countries have documented around 16.8% of the total global NDM-1 producers, the American continent around 10.8%, Africa around 10.8%, and Australia 1.6% [4].

The greatest epidemic threat to the hospital environment is now the strains of Kp, which produce New Delhi metallo-β-lactamases (NDMs) [5,6]. The resistance of Kp New Delhi strains to carbapenems is the result of an enzyme coded by the *bla*NDM gene, which is capable of hydrolyzing almost all β-lactam antibiotics [5,7]. The genes encoding NDMs are often located on plasmids, particularly IncX3, IncFII, IncA/C, FIA/B, and transposon types, and can thus be transferred between bacterial strains by vertical and horizontal transfer. Additionally, mobile genetic elements such as insertion sequences (ISs) can change their location inside the genome, pick up adjacent sequences, and mobilize resistance genes to move (ISCR27), or take part in the expression of carbapenemase genes (ISAba125) [8,9,10].

The metallo-β-lactamase NDM is a lipoprotein anchored to the bacterial outer membrane. This increases the stability of the enzyme, and its secretion in outer membrane vesicles protects against the action of β-lactam antibiotics [11]. It is also conducive to rapid spreading of both the bacteria themselves and the transfer of plasmids carrying the *bla*NDM gene [12].

Fuursted et al. have described the virulence potential of Kp carrying NDM-1 using mouse and nematode models, and suggested a role for this virulence in rapid global spread [13]. Virulence-associated factors (VFs) are associated with different types of infection. They include fimbriae (i.e., 1, 3), siderophores (i.e., enterobactin, salmochelin, aerobactin, yersiniabactin), lipopolysaccharide (LPS) [14,15], and 79 serological K types of capsules [16]. The hypermucoviscosity (HM) phenotype of Kp was initially associated with several specific types of capsules, but it is now known that the HM phenotype is related to the expression of the *rmp*A and *rmpA2* genes and not the capsule type [17,18]. Strain-specific VFs are included in the flexible gene pool, which are responsible for adaptation to a specific replicative niche. The location on the genomic island, genomic islets, phages, plasmids, near integrons, and transposons can all facilitate their horizontal or vertical transfer [14].

Based on seven loci (*mdh, infB, tonB, gapA, phoE, pgi*, and *rpoB*), we classified MDR Kp strains by sequence type (ST). There are 29 recorded NDM variants across several bacterial species [19], which differ in their amino acid substitutions. NDM-1 is the most common variant, and for epidemic (very fast spread, difficult eradication) and clinical (multi-drug resistance, limited therapeutic options) reasons, these strains are a big problem within the hospital environment [3,20]. The most common sequence types of strains of Kp with NDM expression include ST11, ST14, ST15, ST147, ST149, ST231, and ST 258 [21,22,23,24]. Since 2012, there has been a massive expansion of Kp ST11 producing NDM-1 in Poland, which is one of the most remarkable recent phenomena in CPE epidemiology in Europe [20,25]. Since 2016, there has been an uncontrolled spread of these strains in Poland, and their occurrence is now reported in all provinces [26,27]. The multi-resistance of bacteria and the lack of new effective antibiotics reduce the chance of therapeutic success in the treatment of infections. Therefore, prevention is necessary, in particular screening, as well as cutting off transmission pathways of epidemic clumps through hand hygiene and isolation of infected and colonized patients. Scientific research is also needed; that is, detection, analysis, and tracking of the spread of epidemic Kp NDM-1 clones in various regions of Poland.

*K. pneumoniae* contributes to the inflammation of the lungs, liver abscesses, and sepsis, and is an etiological agent of urinary tract infections (UTIs). UTI is the most common infectious complication after renal transplantation (RTx) [28,29]. Most episodes of UTI occur during the first 6 months after transplantation and one of the utmost frequently isolated microorganisms is *Klebsiella* spp. [30,31,32]. Here, we present the phylogenetic relationships and genomic characteristics of Kp NDM isolates from the recent outbreak causing urinary tract infection, and inter alia for renal transplant recipients. We discuss the problem of Kp NDM colonization of the gastrointestinal tract and asymptomatic bacteriuria (ABU) in the context of the risk of UTI, nosocomial infection, and problems with treatment for immunodeficient patients. We used meta-analysis of genomes for genome comparisons of our strains to nationally and internationally recorded strains, exploring their epidemic and evolutionary origins.

## 2. Materials and Methods

### 2.1. Patients and K. pneumoniae Collection

A total of eight Kp NDM isolates were tested, seven of which had been isolated in either 2016 or 2018, from six patients (one patient with two isolates) hospitalized in the Department of Nephrology, Transplantology, and Internal Medicine, the Medical University of Gdańsk, Poland. An eighth isolate was sent for diagnostic purposes to the University Clinical Laboratory in Gdańsk from another hospital in Poland (Toruń) and it was also used as a reference strain for genotyping. The list of strains and their origin is given in Table 1 and Appendix A. The strains were stored at −80 °C (microbank, BioMaxima S.A. Lublin, Poland) for further analysis. The strains were routinely cultured at 37 °C using Lysogeny agar (LA) and broth (LB). 

### 2.2. Antibiotic Susceptibility Testing

Identification and antibiotic susceptibility were determined using the VITEK^®^ 2 Compact (bioMérieux, Marcy l’Étoile, France). Susceptibility assays were interpreted according to EUCAST v 10.0 (2020) recommendations [33]. ESBL production was confirmed using E-test (Etest^®^ strips, bioMérieux, Marcy l’Étoile, France) and carbapenemase production was confirmed with a *β*-*CARBA*^®^ test (Bio-Rad Laboratories, Marnes-la-Coquette, France). The presence of the *bla*_NDM-1_ gene was detected by the PCR technique, as described [34].

### 2.3. Ethical Statements

This study was approved by the Local Bioethics Committee at the Medical University of Gdańsk (Gdańsk, Poland) (decision no. NKBBN/120/2019). The medical personnel responsible for the taking of samples were not involved in authoring the study, and information about patients was not available to the authors of the publication.

### 2.4. DNA Isolation

Genomic DNA was extracted from 3–4 colonies on an LA agar plate using the Genomic DNA Kit (Bioline, Ltd. England), as detailed in the manufacturer’s instructions. The DNA concentrations were measured using a NanoDrop ND-100 (Thermo Fisher Scientific, Spectro-Lab, Łomianki/Warsow, Poland) and were at the level of 10–40 ng μL^−1^.

### 2.5. Clonal Genetic Relatedness Analysis by PCR MP Genotyping

A PCR MP method was used according to the procedure described by Stojowska et al. [35], with the slight modification described by Gołębiewska et al. [36]. Comparisons of electrophoretic profiles, based on band position, were made using Bio-Rad software (FPQuest^TM^ software, BioRad, Ver. 4.5) and UPGMA (unweighted pair-group method with arithmetic averages). The P_i_ cut-off for genotype definition was 96%.

### 2.6. Whole Genome Sequencing (WGS) and Genome Assembling

Genomes of all isolates were sequenced with the Illumina MiSeq platform (Illumina, Wellcome Trust Sanger Centre, Hinxton, UK). Read cleaning (i.e., removing the sequencing adapters), filtration, and trimming according to quality and length criteria were performed using Trimmomatic (Ver. 0.36) [33]. The genomes were de novo assembled using SPAdes (Ver. 3.10.1) [37]. Bacterial sequence reads were assembled into primary contigs, which were deposited in the NCBI’s database with BioProject and BioSamples accession numbers PRJNA688074 and SAMN17168596–603, respectively. Quast (Ver. 5.0.2) [38] was used to generate summary statistics for each assembly (Appendix A).

### 2.7. Phylogenetic Analysis of K. pneumoniae Genomes

The draft genomes were annotated with Prokka (Ver. 1.11) [39] using default parameters, and the pangenome analysis was completed using Roary (Ver. 3.6.0) [40], specifying 95% identity blastp. Using a custom-made pipeline, a multi-FASTA file was obtained for each target core gene. These were aligned gene-by-gene using Muscle (Ver. 3.8.31) [41], then concatenated using a custom python script. The maximum-likelihood (ML) core gene phylogenetic tree was generated from the concatenated alignment core genes using the GTR model in RAxML (randomized accelerated maximum likelihood) (Ver. 8.2.12) [42]. The ggtree R package (Ver. 1.15.6) [43] was used for the visualization, manipulation, and annotation of the phylogenetic trees. The cluster was defined by rhierBAPS [44]. In silico MLST was performed using a BLAST-based tool [45] on de novo genome assemblies. The eight isolates were subjected to phylogenetic analysis against 71 carbapenemase-producing Kp ST11 isolates from Poland, including 67 representatives of the NDM-1 epidemic from 2012–2018 and 4 other known Polish carbapenemase-positive Kp ST11 sequences available from GenBank (accessed on 24 April 2020) [32]. Additionally, data from 101 random Kp genomes publicly available in the PATRIC database (accessed on 24 April 2020) were downloaded for inclusion in the analysis, including 90 Kp isolates belonging to sequence type ST11 and 11 Kp isolates belonging to sequence type ST258, a worldwide dominant carbapenem-resistant Kp clone that is closely related to ST11. Another ST was added in order to give context with other important STs.

A diagram showing the groups of the analyzed strains in this work is presented in Appendix A. Metadata were provided for all bacterial isolates included in the study (Appendix A). These include year, country, and city of isolation; source; specimen type; infection status; clinical information where available (e.g., clinical syndrome, type of infection); patient outcome (e.g., died in hospital, discharged); and mode of acquisition (nosocomial if isolated more than 48 h after admission to hospital, otherwise community acquired).

### 2.8. Single Nucleotide Polymorphism (SNP) Identification

An SNP pairwise difference matrix from the core gene was obtained for ST11 and other STs using snp-dists (Ver. 0.6.3). The SNP analysis of the core genome alignment was completed by aligning the short-reads data from each isolate against the reference, the first outbreak isolate from Poland from 2012 (BioSample accession number SAMN14314560), for a comparison in high resolution among ST11 using Snippy (Ver. 3.1) [46]. The analysis pipeline was as follows: paired-end reads of each strain were mapped to the reference genome, variants were identified and annotated, and the effects of each variant on genes were predicted using the SnpEff tool (Ver. 4.3t) [47].

### 2.9. Virulence Factors and Resistance Gene Analysis

The ABRicate tool was used to identify antimicrobial resistance and virulence genes by running local assemblies. Antimicrobial resistance genes were detected using ResFinder 3.1 [48], with a cut-off of 100% sequence identity. Virulence gene allele sequences were retrieved from the *K. pneumoniae* BIGSdb database at Institut Pasteur [49], with a 95% identity threshold.

### 2.10. K/O Serotypes and Mobile Genetic Element Analysis

Capsule (K) and LPS (O) antigen types were differentiated among the isolates using the Kaptive tool [50]. Plasmid replicon types were identified using PlasmidFinder 2.1. [51]. Comparisons of plasmids were carried out using BLASTn, and the results were visualized using R [52]. The presence of integrative and conjugative elements, ICEKp1 and ICEKp2, was investigated using gene markers described previously [53]. BLAST was used to identify these gene markers in sequence data from our isolates.

### 2.11. Scoring Contigs as Plasmid or Chromosomal from Draft Genomes

The RFplasmid tool [54] was used to predict plasmid and chromosomal contigs from draft assemblies. Therefore, each contig was assigned a plasmid or chromosomal score.

### 2.12. Data Availability

Contig sequences for each genome were deposited in the NCBI’s database, with BioProject and BioSamples accession numbers of PRJNA688074 and SAMN17168596–603, respectively.

## 3. Results

### 3.1. Patients and Samples from the Local Hospital

The seven included Kp NDM cultures were isolated in 2016 and 2018 in the Department of Nephrology, Transplantology, and Internal Medicine, the Medical University of Gdańsk, from six patients with a medical history of kidney disease (Table 1, Appendix A). The eighth isolate was sent for analysis from another hospital in Poland to confirm resistance to carbapenems by molecular methods. It was treated as a reference strain not related to our hospital. Six Kp NDM strains were isolated from the urine of patients with symptoms of UTI (three isolates) or ABU (three isolates), including two renal transplant (RTx) recipients. The other two isolates were from the GI tract (anal swab samples), one from a RTx patient positive for Kp NDM in the urine, and the other from a patient hospitalized at the same time, in the same ward, but without UTI symptoms.

### 3.2. K. pneumoniae Antibiotic Susceptibility Profile

Antibiotic resistance profiles were prepared for each of the Kp NDM isolates (Table 2). ESBL production was confirmed by E-test, and carbapenemase production was confirmed with a β-CARBA^®^ test. By obtaining a 621 bp PCR product, we also confirmed the presence of the *bla*NDM-1 gene in all eight isolates. Most of the tested *Kp* NDM strains showed similar patterns of resistance. All NDM strains were resistant to penicillins, penicillins with β-lactamase inhibitors, all generations of cephalosporins, fluoroquinolones, ertapenem, meropenem, and cefoperazone/sulbactam. In the case of the remaining carbapenems (imipenem), the degree of resistance of the strains varied. Resistance to imipenem was found in four strains, and four strains were also characterized by moderate susceptibility to this drug. We observed large differences in sensitivity to aminoglycosides. None of the strains were resistant to gentamicin while only one strain was resistant, and two strains were moderately sensitive to amikacin. One strain of NDM showed resistance to tobramycin. Most strains remained resistant to trimethoprim/sulfamethoxazole, except for strain number ND32, isolated in 2016. In the analyzed material, one strain was also found to be moderately sensitive to tigecycline. All strains remained sensitive to colistin.

### 3.3. Clonality of Strains from a Local Outbreak

The main aim of this task was to determine whether the strains isolated from kidney disease patients and kidney recipients were from the community or were hospital-acquired. Owing to the emergence of NDM-producing Kp in January 2018 at the Department of Nephrology, Transplantology, and Internal Diseases of the Medical University of Gdańsk, the patients had been monitored for infection and colonization during the period from January to April 2018.

Genotyping of seven Kp strains isolated from six patients, and a reference Kp strain (from another hospital, used as an outlier control), revealed three distinct genotypes, A–C (Figure 1). The Dice similarity coefficient was 96%.

The strain from 2016 (ND32) and the first strain from 2018 (ND24) had unique genotypes, A and B, respectively, indicating a new strain being present in the clinic’s renal ward in 2018. The other strains from 2018 were assigned a third, ‘C’, genotype, and were not consistent with the first strain isolated in 2018. For one patient (P4) from which two isolates were taken (ND88 and ND34), two distinct subgenotypes (C1 and C2) were described. One isolate came from the urine and the other from the patient’s GI tract (anal swab samples). The strains from the local outbreak (ND33) and the reference strain (ND27/Ref) were characterized by a high degree of similarity (98%), and were assigned to a subgenotype, C3.

### 3.4. Whole Genome Sequence Analysis

#### 3.4.1. Phylogenetic Analysis

The seven studied isolates were subjected to phylogenetic analysis against the non-outbreak reference isolate sequenced by us, and 71 publicly available Polish Kp genomes, 67 of which were NDM-1 Tn125A isolates; 4 other known Polish carbapenemase-positive Kp ST11 strains, as described previously [32]; 90 randomly selected Kp strains with sequence type ST11; and 11 randomly selected Kp strains with sequence type ST258 available from the PATRIC database (Figure 2). The core-gene phylogenetic tree for eight isolates from this study is presented in Appendix A.

Phylogenetic analysis with concatenated alignment core genes showed that all seven examined isolates from our hospital represented the ST11 sequence type, with high homogeneity. The seven studied isolates were included in one clade, with 68 Polish isolates (all NDM-1) and 21 isolates from different countries around the world, 10 of which had the *bla*_NDM-1_ gene. These 21 isolates were collected in 2013–2016 in Greece (*n* = 9, all NDM-1), Slovakia (*n* = 5), the Czech Republic (*n* = 3), India (*n* = 2), Spain (*n* = 1), and USA (*n* = 1, NDM-1) (mainly from the EuSCAPE study). Five Kp NDM-1 sequences from the public database were not included in this clade. These isolates were collected in 2012–2013 in Pakistan (*n* = 2), Turkey (*n* = 1), Norway (*n* = 1), and Romania (*n* = 1). Two Kp NDM-1 sequences from Izdebski et al. [32] were also not included in this clade.

#### 3.4.2. SNP Diversity of Core Genomes

Analysis of the cgSNP (core genome SNP) in relation to the reference strain (BioSample accession number SAMN14314560) revealed 7–88 SNPs between the individual Polish isolates sampled in this study. The timing of individual isolates from the outbreak was reflected by a gradual reduction in the number of SNP differences between temporally closer epidemic isolates, from 88 SNP differences between the January 2018 isolate and the April 2018 isolate, to 16 SNP differences between the two April 2018 isolates, showing the process of population diversification.

The isolates collected during the study showed the fewest differences to strains from Poland and Greece; differing from Greece isolates by 46–79 SNPs. Isolate no. ND32 (SAMN17168599), the first isolate from the ward, showed the fewest differences compared with strains from Greece, differing from them by 46–56 SNPs. The analysis showed that isolate no. ND27 (SAMN17168596), isolated at the same time (May 2018) as the outbreak strains, but from a different hospital, differed from the outbreak strains by 16–86 SNPs. Similarly, in the SNPs’ pairwise distance from core gene analysis, a Polish isolate taken from the public database (No. KN21460, 2013) was the closest to the studied isolates, differing from the outbreak strains by 43–64 SNPs and from strains originating in Greece by 25–36 SNPs (Appendix A). The SNP distribution divided the isolates in this homogeneous clade into groups clustering with Polish, Greek, and other international isolates.

#### 3.4.3. Resistomes

Eleven antimicrobial resistance genes in total were detected in the seven isolates from the hospital in Gdańsk, which could be used to split the strains into seven different gene profiles (Table 3 and Appendix A). The resistomes comprised 5 to 10 acquired genes and an average isolate had 7.6 of these.

A phenotype of resistance to β-lactams was generated from the *bla*NDM-1, *bla*CTX-M-15, *bla*OXA-1, and *bla*TEM-1 genes. All four genes were present in the ND34 (GpC2)- and ND88 (GpC1)-related isolates; three genes (*bla*NDM-1, *bla*CTX-M-15, *bla*OXA-1) in ND32 (GpA), ND111 (GpC), ND96 (GpC), and ND27 (ref) (GpC3); and only two (*bla*NDM-1, *bla*CTX-M-15) in ND24 (GpB) and ND33 (GpC3). The *aac (6′)-Ib* gene, which is responsible for aminoglycoside resistance, was detected in seven isolates. The *aad*A1 gene was only detected in isolate ND33. Resistance to phenicols was encoded by the *cat*1 and *cat*4 genes for ND111 and ND88, and for other isolates by the *cat*A1gene only. The ND24 isolate lacked resistance genes for this group of antibiotics. The *sul1* gene, responsible for sulfonamide resistance, was included in all isolates. One or both of the *dfr*A1/*dfr*A14 genes, which contribute to resistance to trimethoprim, were detected in all isolates. Most of the genes were determined to be plasmidic. Overall, these genes enabled us to specify a perfect phenotype to genotype correlation for antibiotic susceptibility data.

#### 3.4.4. Plasmid Analysis

The previous Polish outbreak study revealed a prevailing presence of the IncFII/pKPX-1 and IncR replicons, and high differentiation of the species. This study provides new data on the plasmid content of the epidemic genotypes resulting from a local outbreak of a nationally distributed clone. Eight replicons in total were detected across the seven isolates, which could be split into four profiles of one to six replicons each (Table 2 and Table 3), again showing the heterogeneity of the organism. The dominant replicon was IncFII/pKPX-1, which was present in six isolates.

#### 3.4.5. Mobile Genetic Elements Analysis

The frequency of integrative conjugation elements (ICEs) ICEKp1 and ICEKp2 in clinical Kp isolates was determined. ICEKp1 was identified in 113 (70.2%) international strains and all eight strains tested in this study. ICEKp2 was present in 159 out of 161 (98.8%) international genomes, and in all genomes sequenced by us. Its presence was not correlated with ICEKp1. Among the Kp NDM-1 strains, both ICEKp1 and ICEKp were identified in 89 (96.7%) strains. The presence of ICEKp2 correlated with the presence of ICEKp1 for Kp NDM-1 strains. Additionally, the presence of two closely related sequence types, as demonstrated by the genome analysis, suggests a stable inheritance of ICEKp2. ICE contributes to the spread of antimicrobial multi-resistance genes and pathogenicity [55].

Among the eight studied strains of Kp NDM-1, ICEKp2 did not coexist with the plasmid carrying the *rmp*A and *rmp*A2 genes encoding the so-called ‘mucus factor’, which are associated with the overexpression of proteins in strains with the K1 or K2 capsule type, and increased virulence [55,56]. In contrast, the seven international strains of Kp carrying the plasmid including the *rmp*A and *rmp*A2 genes always had ICEKp2 and a partial (*n* = 5) or complete (*n* = 2) ICEKp1 element (Appendix A, Appendix A).

#### 3.4.6. Serotypes

The K and O antigens of the outbreak isolates were identified as KL24 and O2 variant 1 (O2v1), respectively, with no differences between the Polish ST11 NDM-1 strains and some of the international ST11 NDM-1 strains, especially from Greece (Table 4 and Appendix A). The within-study serotype diversity of the ST11 isolates and control ST258 isolates used was significant, corresponding to previous reports.

#### 3.4.7. Virulomes

Virulence-associated gene analysis (Table 4 and Appendix A) showed an identical arrangement for all strains, including the chromosomal *fim*A-K operon, responsible for the type 1 fimbriae involved in adhesion, and the chromosomal *mrk*ABCDF and *mrk*HIJ operons, responsible for the type 3 fimbriae, involved in adhesion and biofilm formation. Additional groups of virulence genes considered included those for iron secretion systems: the *fyu*A, *irp*1/2, and *ybt*AEPQSTUX genes specifying synthesis and secretion of yersiniabactin; *iro*E specifying synthesis of salmochelin; and the *ybd*A, *ent*ABCEF, *fes*, and *fep*ABCDG genes specifying synthesis and secretion of enterobactin. The same virulence gene profile was found in close relatives of the outbreak isolates from Poland, Greece, Romania, Norway, and Turkey.

## 4. Discussion

*Klebsiella pneumoniae* (Kp) is responsible for many nosocomial infections, especially in immunocompromised patients [36,57]. Hospital wards with immunocompromised or immunosuppressed patients have reported more frequent outbreaks of Kp [58]. The Kp is characterized by a high epidemic risk in hospitals owing to increasingly common multi-drug resistance and easy acquisition of fitness genes and various virulence factors. These bacteria have the ability to create a biofilm, which makes it easier for them to colonize various ecological niches, and at the same time hinders eradication.

In some patients, colonization is associated with carrier status for multi-drug-resistant strains. UTIs, ABU, and colonization/carriage are a serious problem as they are mainly responsible for the global expansion of carbapenemase-producing Kp clones [59].

Among our patients were RTx recipients with symptomatic UTI, patients with asymptomatic bacteriuria, and patients with GI tract colonization. All these patients had a history of either chronic or kidney failure or acute kidney injury and were hospitalized in one hospital, and from each of them, Kp NDM-1 was isolated.

In patients with kidney failure, including RTx recipients, the choice of antibiotics can be limited by impaired renal function and the possibility of interaction with other drugs; it is very difficult to determine an optimal prophylaxis [60]. Many antibiotics prove ineffective owing to the lack of a capture point in both structurally and functionally altered cells. Owing to the increasing, and sometimes inappropriate, use of antibiotics, especially in this group of patients, many reports emphasize the growing incidence of multi-drug-resistant strains, in particular those from the Enterobacteriaceae family [60,61].

In recurrent UTIs, the probability that a carbapenem-resistant strain will be the etiological factor increases with each subsequent episode [62]. The use of amoxicillin with clavulanic acid, metronidazole, clindamycin, or piperacillin with tazobactam increases the intensity of gastrointestinal colonization with Kp ESBL (+). UTIs or ABU can be caused by the same Kp bacteria that colonize the patient’s digestive system. Urinary tract infections caused by ESBL strains require treatment with carbapenems, and that therapy often generates carbapenem-resistant Kp.

The causes of ABU can be explained in two ways. On one hand, uropathogenic strains are less virulent, or some genes are not expressed because of mutations (deletions, insertions). According to Zdziarski et al., asymptomatic bacteriuria is a result of genome reduction and inactivation of genes encoding virulence-associated factors by the accumulation of point mutations or deletions [63]. Attenuation of virulence may also occur among isolates during long-term carriage in the urinary tract [64]. On the other hand, the innate immunity response may be too weak to cause symptoms. ABU favors the acquisition of resistance because bacteria colonize the urinary tract, forming a biofilm, which is a perfect environment for horizontal gene transfer, similar to the GI tract.

In recent years, there have been numerous reports of epidemic outbreaks caused by MDR *K. pneumoniae* strains [65,66,67]. The first stage of our research was to assess the epidemic situation in the Department of Nephrology, Transplantology, and Internal Medicine, considering either commensal strains carried by the patient or strains of hospital origin. In a patient who stays in the hospital environment for a long time, the commensal intestinal microbiota can be replaced by the hospital microbiota. The Kp NDM strains showed a high degree of genetic similarity (GpC) among subgenotypes (Cp1–Cp3), distinguishing them from the 2016 strain (GpA) and the first strain that appeared in 2018 (GpB) (Figure 1). Our starting hypothesis that an endemic GpA strain could be responsible for the epidemic situation was thus rejected. Isolate no. ND24 (no. 2), with the unrelated genotype GpB, was probably a community-acquired or commensal strain associated with the microbiota of a patient. High genetic similarity of the DNA patterns for other isolates could indicate a clonal origin and local epidemic situation.

All strains showed an identical or similar resistance profile. Only for imipenem, amikacin, and tigecycline were differences in antibiotic resistance noticeable. These differences were also reflected in the genotypes and sub-genotypes of these isolates. These slight differences in genotypes and resistance profiles are associated with the progression of the isolation timeline of the Kp NDM-1 strains. To answer the question of how much variability exists at the genomic level among Kp NDM strains, we used WGS to determine the level of genetic relatedness among our isolates and between these and other strains from worldwide sources.

We have shown that our carbapenem-resistant isolates belong to the NDM-1 variant. The NDM-1 variant is considered a pandemic and has been recorded in as many as 86 countries around the world, including in Poland [68,69,70].

Phylogenetic analysis with cgMLST indicated high homogeneity with respect to sequence type (ST). All of our isolates belong to the lineage ST11, whose role in the hospital environment is significant. It has been shown that elements of ST11, together with ST340, participated in the evolution of the clonal complex CC258, which has global dissemination [71,72]. Kp core-genome phylogenetic tree analysis confirmed that these strains, belonging to one big clade in Poland [32], have a high similarity to European isolates from Greece, Slovakia, the Czech Republic, Spain, and other more distant countries such as India and the USA (Figure 2).

Analysis of the cgSNP profiles indicated 7–88 SNP differences between the individual Polish isolates. The first isolate that appeared in 2018 on the ward was the least different from Greek strains, and the isolates reveal the phenomenon of an increase in SNP differences from Greek strains over the course of the epidemic situation, in line with the diversification of the population. Although related Kp ST11 strains belonging to the same basal clade occurred globally (Appendix A), the analyzed genotype appeared mainly in Europe, especially in South-Central Europe. This study demonstrates the international character of these sublines, with documented epidemic potential in Poland and Greece.

The rapid spread of Kp NDM strains to all continents is largely related to the global development of tourism [71]. Therefore, microbiological testing of patients coming back from travel, immediately after their hospital admission, is extremely important, especially if they were hospitalized at that time in countries with endemic NDM-positive strains [72,73]. The reasons for dissemination of Kp NDM should also be looked at among medical personnel coming back from medical missions in other countries with a high rate of Kp NDM incidence.

The virulence properties of Kp have been documented in many reports [36,74,75]. Capsule, LPS, siderophores, type 1 and 3 fimbriae, and biofilm formation are all subjects of recent pathogenesis research. Whole-genome sequencing (WGS) is the best molecular tool for examination of the capsular locus and the population structure of Kp. The capsule is essential for infection and protection against the host immune system. Based on association with capsule lineages, strain typing can be performed more efficiently than with MLST [50].

WGS analysis showed the same virulence profile of genes for all isolates, regardless of each sample’s origin (ABU/UTI/GI tract). Based on the type of K24 capsule, LPS o2v1 antigen, and type 1 and 3 fimbriae were detected for our isolates. The K24 capsule type is found in Polish ST11 NDM-1 strains and some strains from Greece (Figure 2). Many observations have confirmed the roles of type 1 fimbriae and type 3 fimbriae in biofilm formation, especially during chronic infections [76,77,78]. Biofilm protects against the effects of many antibiotics and promotes the growth of resistance to antibiotics. Biofilm is also produced in the GI tract, and creates an environmental niche in which different populations of bacteria come into contact with each other and can exchange genes, determining resistance to antibiotics and virulence.

Kp acquire iron through a system based on genes encoding siderophores such as yersiniabactin, salmochelin, and enterobactin, enabling the high virulence of multi-drug-resistant Kp NDM strains. The presence of the yersiniabactin high-pathogenicity island contributes to high virulence in our isolates. This type of siderophore is mainly characteristic of Kp strains responsible for the colonization of the respiratory system [79,80,81], but has also been reported in strains isolated from UTI patients [82,83]. In the case of ABU, the expression of yersiniabactin with the presence of an efficiently forming biofilm was confirmed in human urine [82]. The environment of the urinary tract is iron-poor, and bacteria must thus compete for iron. The adoption of an additional iron uptake system that is able to avoid sequestration by lipocalin-2 is thus beneficial for pathogenic bacteria. Yersiniabactin was detected among Kp NDM-1 strains isolated from Poland, Greece, Romania, Norway, and Turkey. Salmochelin and enterobactin, but not aerobactin, have additionally been implicated in Kp NDM-1 virulence.

Based on WGS, we found no differences in virulence profile across the strains responsible for UTI or ABU, and for isolates colonizing the gastrointestinal tract. The virulence of Kp probably does not determine the appearance or absence of symptoms of the disease. However, further research is required to examine strains from a larger population.

Besides clonal transmission, mobile genetic elements, including transposons and plasmids, play a role in the dissemination of multidrug-resistant clones of Kp. The *bla*NDM-1, *bla*KPC-2, *bla*VIM, *bla*IMP, and *bla*OXA-48 genes are responsible for resistance to carbapenems, and are transferred to other microorganisms via horizontal gene transfer [84]. The co-existence of several unrelated plasmids carrying the *bla*NDM-1gene and other genes encoding resistance to carbapenems, e.g., *bla*OXA-48, has been reported in the same bacterial cell [85]. Additionally, other antibiotic-resistant markers were detected in Kp, classifying them as an alert pathogen [6,86].

In our research, Kp NDM-1 isolates carrying the *bla*NDN-1gene were also carriers of genes related to resistance to antibiotics belonging to different classes: β-lactams (*blaOXA-1, blaCTX-15, blaTEM*), aminoglycosides (*Aac* (6′)-*Ib*, *aadA1*), phenicols (*cat1, cat4, catA1*), sulfonamides (*sul1*), and trimethoprim (*dfr*A1/*dfr*A14). Various types of plasmids such as IncR, IncF, IncA/C, and IncNIncL/M are involved in the carriage of the *bla*NDM-1 gene [87]. Concerning plasmids and replicons, our isolates showed heterogeneity (Table 3) and such profiles have not been detected previously. The strains that showed different genotypes from those identified in previous epidemiological studies also carried different sets of plasmid replicons from those strains considered epidemically related. The epidemic strains were characterized by rich plasmid replicon profiles: FIB (K), FII (K), FII (pKPX1), HI1A (NDM-CIT), and HI1B (pNDM-CIT) or FIB (K), FII (K), FII (pKPX1), HI1A (NDM-CIT), HI1B (pNDM-CIT), and X1, while the strain isolated in 2016 carried only the FII (pKPX1) plasmid replicon. Only two plasmid replicons, FII (Yp) and R, were detected for the first Kp NDM-1 strain from 2018. This is further evidence of a different origin of these strains.

Mobile genetic elements, e.g., integrative conjugation elements (ICEs), are widely recognized as contributing to the spread of antimicrobial resistance and virulence genes included in islands of pathogenicity. ICEKp1 and ICEKp2 can co-exist in Kp, and were documented during our global analysis of Kp NDM-1. ICEKp2 is often detected in the presence of a plasmid carrying the *rmp*A and *rmp*A2 genes, which is typically associated with Kp (hvKp) strains with a hypermucoviscous phenotype [55,56]. Despite the presence of the ICEK2 plasmid, our strains did not show co-existence of this plasmid. The presence of several plasmids in the same strain carrying genes related to antibiotic resistance and high-pathogenicity islands makes Kp dangerous both in hospital environments and outside of them [88].

## 5. Conclusions

In the case of renal transplant recipients, UTI is common because long-term immunosuppression is associated with many side effects, particularly an increased risk of infection. Organ recipients are subjected to preventive antibiotic therapy before and after transplant surgery, in order to minimize the risk of complications. Antibiotic therapy may alter the urinary microbiome composition and enrich for uropathogenic species [68]. For chronic recurrent UTI, repeated antibiotic treatment is required, hence the accumulation of antibiotic resistance in uropathogenic Kp strains.

In addition, this group of patients is particularly exposed to hospital infections; therefore, it was necessary to analyze the epidemic situation in the hospital ward.

Asymptomatic bacteriuria is also a significant problem. We ask ourselves whether or not to treat asymptomatic bacteriuria, especially considering that treatment of asymptomatic bacteriuria with targeted or empiric antibiotic therapy leads to the formation of multi-drug-resistant strains, e.g., producing carbapenemase.

Researchers are constantly looking for differences in strains that cause symptoms of UTI and those associated only with the carrier and ABU. Hence our interest in Kp NDM strains isolated not only from UTI, but also from ABU and GI tract colonization.

The aim of the study was the phenotypic and genotypic characterization of Kp strains producing metallo-β lactamase NDM, isolated from patients with kidney diseases and RTx recipients. We used WGS analysis to assess the genome diversity of local isolates in relation to national and global strains of Kp NDM-1 with the sequence type ST11. The correlation between strains was determined using PCR MP genotyping, confirming the presence of local outbreak and strains unrelated to the epidemic situation. Mobile genetic elements, in particular, integrative conjugative elements, are responsible for the dissemination of multi-drug resistance and virulence. Meta-analysis of genomes did not show signatures of UTI- and ABU-associated strains with high antimicrobial resistance. Colonization and immunosuppression can be additional risk factors, although this is seldom reported. In immunocompromised patients or those treated with antibiotics for prevention, the gut or ABU can act as reservoirs for the transmission of Kp NDM in the hospital setting.

## Figures and Tables

**Figure 1 genes-12-01285-f001:**
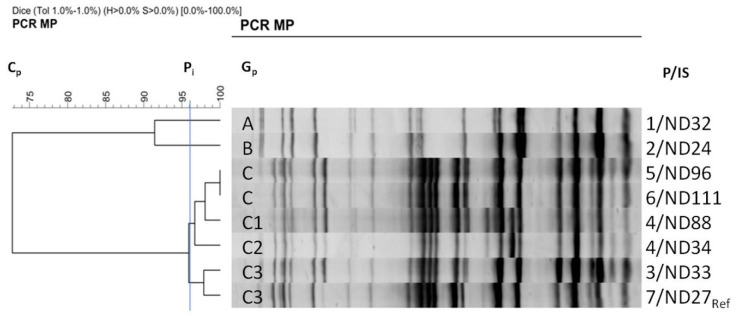
Genetic relatedness of 7 *K. pneumoniae* NDM-1 strains isolated from six patients and a reference *K. pneumoniae* NDM-1 strain (from another hospital used as a control) based on the PCR MP method. A dendrogram of similarity of the genetic profiles of the tested strains was developed using FPQuest Software version 4.5 (BIO-RAD) (Dice, UPGMA). Cp—the complete similarity equal to 73%; Pi—the level of identity equal to 96%; unique genotypes (Gp) are marked with the letters **A**–**C**; IS—isolate; P—patient.

**Figure 2 genes-12-01285-f002:**
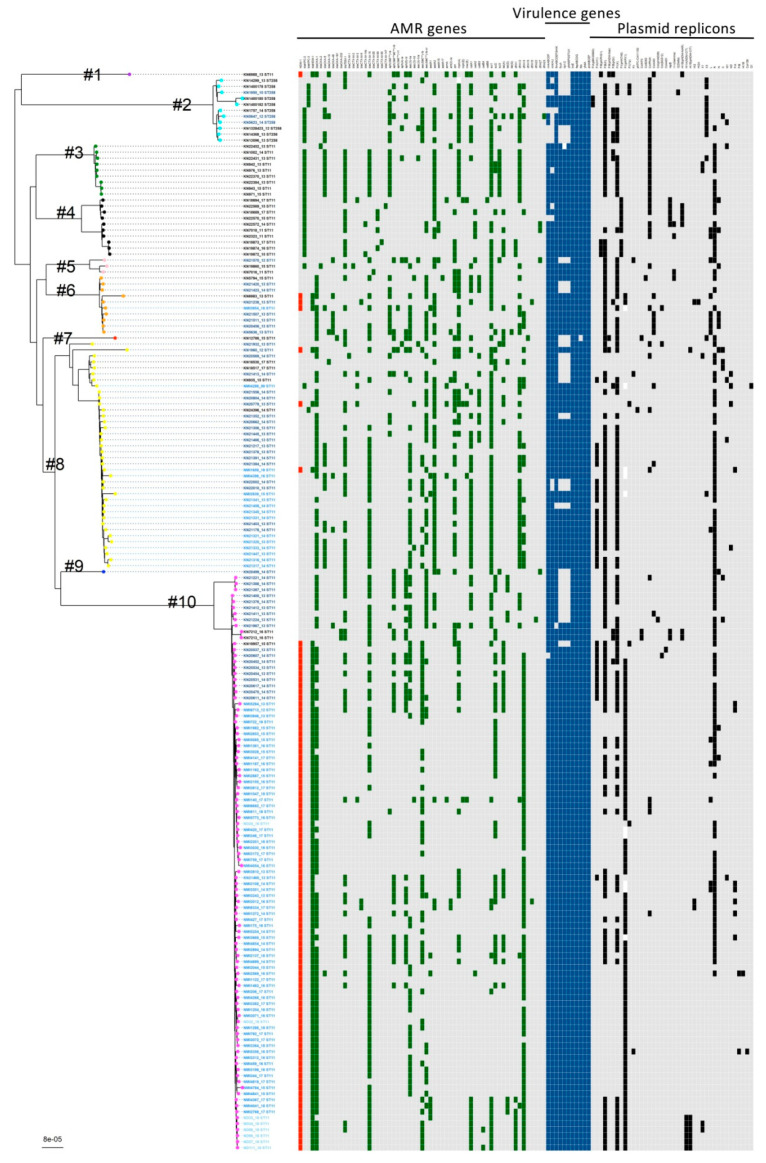
*K. pneumoniae* core-gene phylogenetic tree. A maximum likelihood phylogenetic tree was constructed using 4014 core genes with the genome sequences of 180 *K. pneumoniae* isolates. The cluster numbers (#1–#10) are labelled on the phylogenetic tree and the colour of the circle in the external node is linked to the cluster (purple—cluster no. 1, cyan—cluster no. 2, green—cluster no. 3, black—cluster no. 4, pink—cluster no. 5, orange—cluster no. 6, red—cluster no. 7, yellow—cluster no. 8, blue—cluster no. 9, magenta—cluster no. 10). The origin of the isolates is distinguished by font colours in the tree: light blue—isolates sequenced in this work; blue—from Poland; deep blue—from Europe; black—from other countries. The sequence type (ST) is indicated for each isolate, following the isolate name. Numbers after the isolate name correspond to original numbers of the study isolates or GenBank assembly numbers. In the heatmap, the presence of the *bla*_NDM-1_ gene is indicated by red (present) or gray (absent) and the presence/absence profile of the genotype for 60 genes encoding antimicrobial resistance (green—present, gray—absent), 11 genes encoding virulence determinants (blue—present, gray—absent), and 40 plasmid replicons is indicated.

**Table 1 genes-12-01285-t001:** Overview of patient and strain details used in the epidemiological analysis.

Patients	Isolates	Collection Number	Samples	#Date of Isolation	Result	Renal Transplantation (RTx)
1	1	ND32	urine	2016	UTI	no
2	2	ND24	urine	2018	ABU	yes
3	3	ND33	anal swab	2018	Colonization/carriage	no
4	4	ND34	anal swab	2018	Colonization/carriage	yes
	5	ND88	urine	2018	ABU	yes
5	6	ND96	urine	2018	UTI	no
6	7	ND111	urine	2018	ABU	no
7	8 *	ND27	urine	2018	UTI	no

Legend: * strain used as a reference from another hospital; # in order of isolation; UTI—urinary tract infection; ABU—asymptomatic bacteriuria.

**Table 2 genes-12-01285-t002:** Antibiotic susceptibility profiles for the tested *K. pneumoniae* strains.

Antibiotic	Abbreviation	Name of Isolate *K. pneumoniae* New Delhi							
		ND32	ND24	ND33	ND34	ND88	ND96	ND111	ND27
Ampicillin	AMP	R	R	R	R	R	R	R	R
Amoxicillin/Clavulanic acid	AMC	R	R	R	R	R	R	R	R
Piperacillin/Tazobactam	TZP	R	R	R	R	R	R	R	R
Cephalotin	CEF	R	R	R	R	R	R	R	R
Cefuroxime	CXM	R	R	R	R	R	R	R	R
Cefotaxime	CTX	R	R	NDt	R	R	R	R	R
Ceftazidime	CAZ	R	R	R	R	R	R	R	R
Cefepime	FEP	R	R	R	R	R	R	R	R
Cefoperazone/Subactam	SUL	R	R	R	R	R	R	R	R
Ertapenem	ETP	R	R	R	R	R	R	R	R
Imipenem	IPM	I	R	R	I	I	R	I	R
Meropenem	MEM	R	R	R	R	R	R	R	R
Amikacin	AMK	I	R	S	S	I	S	S	S
Gentamicin	GEN	NDt	S	S	NDt	NDt	NDt	NDt	NDt
Tobramycin	TOB	NDt	R	S	NDt	NDt	NDt	NDt	NDt
Tigecycline	TGC	NDt	I	S	NDt	NDt	NDt	NDt	NDt
Ciprofloxacin	CIP	R	R	R	R	R	R	R	R
Trimethoprim/Sulfamethoxazole	SXT	S	R	R	R	R	R	R	R
Colistin	CST	NDt	S	S	NDt	S	NDt	S	NDt

Legend: S, susceptible—orange; R, resistance—blue; I, intermediate—yellow; NDt, not determined—grey.

**Table 3 genes-12-01285-t003:** *K. pneumoniae* NDM-1 ST11 isolates included in this study and from the PATRIC database—plasmid replicon profile and resistomes based on sequencing WGS.

Isolate Name	Plasmid Replicon Profile	Acquired Antimicrobial Resistance Genes ^a^							
		β-Lactams	Aminoglycosides ^b^	Fluoroquinolones ^b^	Macrolide, Linosamide, Streptogramin Antibiotics	Phenicols	Sulphonamides	Tetracycline	Trimethoprim
ND32	FII (pKPX1)	*bla*_NDM-1_, *bla*_CTX-M-15,_*bla*_OXA-1_	*aac (6′)-Ib*	-	-	*-*	*-*	-	*dfrA14*
ND111	FIB (K), FII (K), FII (pKPX1), HI1A (NDM-CIT), HI1B (pNDM-CIT)	*bla*_NDM-1_, *bla*_CTX-M-15_, *bla*_OXA-1_	*aac (6′)-Ib*	-	-	*catA1, catB4*	*sul1*	-	*dfrA1, dfrA14*
ND27	FIB (K), FII (K), FII (pKPX1), HI1A (NDM-CIT), HI1B (pNDM-CIT)	*bla*_NDM-1_, *bla*_CTX-M-15_, *bla*_OXA-1_	*aac (6′)-Ib*	-	-	*catA1*	*sul1*	-	*dfrA1, dfrA14*
ND96	FIB (K), FII (K), FII (pKPX1), HI1A (NDM-CIT), HI1B (pNDM-CIT)	*bla*_NDM-1_, *bla*_CTX-M-15_, *bla*_OXA-1_	*aac (6′)-Ib*	-	-	*catA1*	*sul1*	-	*dfrA1, dfrA14*
ND88	FIB (K), FII (K), FII (pKPX1), HI1A (NDM-CIT), HI1B (pNDM-CIT), X1	*bla*_NDM-1_, *bla*_CTX-M-15_, *bla*_OXA-1_, *bla*_TEM-1_	*aac (6′)-Ib*	-	-	*catA1, catB4*	*sul1*	-	*dfrA1, dfrA14*
ND24	FII (Yp), R	*bla*_NDM-1_, *bla*_CTX-M-15_	*aac (6′)-Ib*	-	-	*-*	*sul2*	-	*dfrA14*
ND33	FIB (K), FII (K), FII (pKPX1), HI1A (NDM-CIT), HI1B (pNDM-CIT)	*bla*_NDM-1_, *bla*_CTX-M-15_	*aadA1*	-	-	*catA1*	*sul1*	-	*dfrA1, dfrA14*
ND34	FIB (K), FII (K), FII (pKPX1), HI1A (NDM-CIT), HI1B (pNDM-CIT), X1	*bla*_NDM-1_, *bla*_CTX-M-15_, *bla*_OXA-1_, *bla*_TEM-1_	*aac (6′)-Ib*	-	-	*catA1*	*sul1*	-	*dfrA1, dfrA14*
KN1960	C, FIB (K), FIB (pNDM-Mar), FII (K), HI1B (pNDM-MAR), N	*bla*_NDM-1_, *bla*_CTX-M-15_, *bla*_OXA-1_	*aac (6′)-Ib, aac (3)-IIa, aph (3′′)-Ib, aph (3′)-VI, aph (6)-Id*	*qnrS1*	*mph* (A)	*catA1*	*sul1, sul2*	-	*dfrA14*
KN20454	FIA (HI1), FIB (K), FII (K), FII (pKPX1), R	*bla*_NDM-1_, *bla*_CTX-M-15_, *bla*_OXA-1_, *bla*_TEM-1_	*aac (6′)-Ib, aac (3)-IIa, aph (3′′)-Ib, aph (6)-Id*	*-*	*mph* (A)	*-*	*sul2*	*tet* (A)	*dfrA14*
KN20534	FIA (HI1), FIB (K), FII (pKPX1), R	*bla* _NDM-1_	*aac (6’)-Ib*	*-*	-	*-*	*-*	-	*dfrA14*
KN20537	Col440II, FIA (HI1), FIB (K), FII (K)	*bla*_NDM-1_, *bla*_OXA-1_	*aac (6′)-Ib, aac (3)-IIa*	*-*	*mph* (A)	*-*	*-*	*tet* (A)	*-*
KN20779	FII (Yp), M2, R	*bla*_NDM-1_, *bla*_CTX-M-3_, *bla*_OXA-1_, *bla*_TEM-1_	*aac (6′)-Ib-cr, aac (3)-IId, aadA2, armA*	*aac (6′)-Ib-cr, qnrB4*	*mph* (A), *mph* (E), *msr* (E)	*catB3*	*sul1*	*tet* (A)	*dfrA12*
KN21238	FIB (K) (pCAV1099-114), FIB (pQiI), FII (pKPX1), HI2, HI2A	*bla*_NDM-1_, *bla*_CTX-M-9_	*aadA2*	*-*	-	*catA1*	*sul1*	-	*dfrA16*
KN21460	FIA (HI1), FIB (K), FII (K), M1, R	*bla*_NDM-1_, *bla*_TEM-1_	*aac (6′)-Ib, aph (3′′)-Ib, aph (6)-Id*	*-*	*mph* (A)	*-*	*sul2*	*tet* (A)	*dfrA14*
KN6983	Col440I, FIB (K), FII (K), FII (pKPX1), L	*bla*_NDM-1_, *bla*_CTX-M-15_, *bla*_OXA-1,-48_	*aac (6′)-Ib-cr, aac (3)-IIa, aph (3′)-Ia, aadA2*	*aac (6′)-Ib-cr*	*mph* (A)	*-*	*sul1*	-	*dfrA12*
KN6988	Col440l, FIB (pKPHS1), FIB (pNDM-Mar), FIB (pQil), HI1B (pNDM-MAR)	*bla*_NDM-1_, *bla*_CTX-M-15_, *bla*_OXA-1,-10_, *bla*_TEM-1_	*aac (6′)-Ib-cr, aac (3)-IIa, aph (3′′)-Ib, aph (3′)-VI, aph (6)-Id, armA*	*aac (6′)-Ib-cr, qnrS1*	*msr* (E)	*catA1*	*sul1*	*tet* (A)	*dfrA1*
KN20452	FIA (HI1), FIB (K), FIB (pQiI), FII (K), FII (pKPX1), R	*bla*_NDM-1_, *bla*_CTX-M-15_, *bla*_OXA-1_, *bla*_TEM-1_	*aac (6′)-Ib, aac (6′)-Ib-cr, aac (3)-IIa, aph (3′′)-Ib, aph (6)-Id*	*aac (6′)-Ib-cr*	*mph* (A)	*-*	*sul2*	*tet* (A)	*dfrA14*
KN20470	FIA (HI1), FIB (K), FII (K), FII (pKPX1), R	*bla*_NDM-1_, *bla*_CTX-M-15_, *bla*_OXA-1_, *bla*_TEM-1_	*aac (6′)-Ib, aac (3)-IIa, aph (3′′)-Ib, aph (6)-Id*	*-*	*mph* (A)	*-*	*sul2*	*tet* (A)	*dfrA14*
KN20531	FIA (HI1), FIB (K), FII (K), FII (pKPX1)	*bla*_NDM-1_, *bla*_CTX-M-15_, *bla*_OXA-1_	*aac (6′)-Ib, aac (3)-IIa*	*-*	-	*-*	*-*	*tet* (A)	*dfrA14*
KN20607	Col (BS512), FIA (HI1), FIB (K), FII (K), R	*bla*_NDM-1_, *bla*_CTX-M-15_, *bla*_OXA-1_, *bla*_TEM-1_	*aac (6′)-Ib, aac (3)-IIa, aph (3′′)-Ib, aph (6)-Id*	*-*	*mph* (A)	*-*	*sul2*	*tet* (A)	*dfrA14*
KN20611	FIA (HI1), FIB (K), FII (K), FII (pKPX1), R	*bla*_NDM-1_, *bla*_CTX-M-15_, *bla*_OXA-1_, *bla*_TEM-1_	*aac (6′)-Ib, aac (6′)-Ib-cr, aac (3)-IIa, aph (3′’)-Ib, aph (6)-Id*	*aac (6′)-Ib-cr*	*mph* (A)	*-*	*sul2*	*tet* (A)	*dfrA14*
KN20617	FIA (HI1), FIB (K), FII (K), FII (pKPX1), R	*bla* _NDM-1_	*aac (6′)-Ib, aph (3′′)-Ib, aph (6)-Id*	*-*	*mph* (A)	*-*	*sul2*	*tet* (A)	*dfrA14*
KN19957	FIA (HI1), FIB (K), FII (K), FII (Yp), R	*bla*_NDM-1_, *bla*_CTX-M-15_	*aac (6′)-Ib, aac (3)-IIa*	*qnrB9*	*mph* (A)	*-*	*sul1*	*tet* (A)	.

Legend: ^a^ Only acquired resistance genes are shown, as identified by ResFinder 3.1.; ^b^ the *aac (6′)-Ib-cr* gene shown in the ‘aminoglycosides’ and ‘fluoroquinolones’ columns is the same gene, conferring resistance to both classes of antimicrobials.

**Table 4 genes-12-01285-t004:** *K. pneumoniae* NDM-1 ST11 isolates sequenced in this study and those included from the PATRIC database, serotype data, and virulomes. Legend: 0—absence, 1—presence.

Sample Name	Serotypes		Virulence Factors										
	Capsule (K) Antigen Types	LPS (O) Antigen Types	Type 3 Fimbriae		Type 1 Fimbriae	High-Pathogenicity Island: Yersiniabactin			IroA System: Salmochelin Synthesis	Fep-Ent System: Enterobactin			
			*mrkABCDF*	*mrkHIJ*	*fimABCDEFGHIK*	*fyuA*	*irp1/2*	*ybtAEPQSTUX*	*iroE*	*fepABCDG*	*fes*	*ybdA*	*entABCEF*
ND32	KL24	O2v1	1	1	1	1	1	1	1	1	1	1	1
ND111	KL24	O2v1	1	1	1	1	1	1	1	1	1	1	1
ND27	KL24	O2v1	1	1	1	1	1	1	1	1	1	1	1
ND96	KL24	O2v1	1	1	1	1	1	1	1	1	1	1	1
ND88	KL24	O2v1	1	1	1	1	1	1	1	1	1	1	1
ND24	KL24	O2v1	1	1	1	1	1	1	1	1	1	1	1
ND33	KL24	O2v1	1	1	1	1	1	1	1	1	1	1	1
ND34	KL24	O2v1	1	1	1	1	1	1	1	1	1	1	1
KN1960	KL14	O3b	1	1	1	1	1	1	1	1	1	1	1
KN20454	KL24	O2v1	1	1	1	1	1	1	1	1	1	1	1
KN20534	KL24	O2v1	1	1	1	1	1	1	1	1	1	1	1
KN20537	KL24	O2v1	1	1	1	1	1	1	1	1	1	1	1
KN20779	KL105	O2v2	1	1	1	1	1	1	1	1	1	1	1
KN21238	KL15	O4	1	1	1	1	1	1	1	1	1	1	1
KN21460	KL24	O2v1	1	1	1	1	1	1	1	1	1	1	1
KN6983	KL15	O4	1	1	1	1	1	1	1	1	1	1	1
KN6988	KL103	O2v1	1	1	1	0	0	0	1	1	1	1	1
KN20452	KL24	O2v1	1	1	1	1	1	1	1	1	1	1	1
KN20470	KL24	O2v1	1	1	1	1	1	1	1	1	1	1	1
KN20531	KL24	O2v1	1	1	1	1	1	1	1	1	1	1	1
KN20607	KL24	O2v1	0	1	1	1	1	1	1	1	1	1	1
KN20611	KL24	O2v1	1	1	1	1	1	1	1	1	1	1	1
KN20617	KL24	O2v1	1	1	1	1	1	1	1	1	1	1	1
KN19957	KL24	O2v1	1	1	1	0	0	0	1	1	1	1	1

## Data Availability

Contig sequences for each genome were deposited in the NCBI’s database with BioProject and BioSamples accession numbers PRJNA688074 and SAMN17168596–603, respectively.

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
