# Peer review of "Genetic Background and Antibiotic Resistance Profiles of K. pneumoniae NDM-1 Strains Isolated from UTI, ABU, and the GI Tract, from One Hospital in Poland, in Relation to Strains Nationally and Worldwide"

_genes, 2021, doi:10.3390/genes12081285_

Round 1

Reviewer 1 Report

Major concerns:

  • Concept errors: Meta analysis of genomes was used for genome comparisons, not metagenome analysis. The latter is usually used for microbiome, a mixture of genomes. SAMN14314560 (Line 183, or Line 298) is a BioSample accession number in BioSample database, not a GenBank accession number used in nucleotide sequences database.
  • The aim was to conduct phenotypic and genotypic characterization, as the authors stated. However, the tables and figures in the manuscript did not clearly reflect this. Suggest moving Table S3, the phenotypic antibiotic resistance spectrum, to the manuscript, which is more important to the readers; moving Fig. 3 to supplemental materials, conjugation elements of which are less important; simplifying Tables 2 & 3 to focus on the topic with reduction of replicated info and other isolates.
  • For a comparison of PCR MP and WGS, show WGS phylogeny tree along with PCR MP results in Fig. 1. The relationships between 8 isolates will be much clearer than those shown in Fig. 2. Current Fig. 2 is difficult to readers. The authors may consider simplifying it or selecting representative isolates to reduce the number, leaving Fig. 2 to supplemental materials.
  • It will be interesting to see the difference between cgMLST phylogeny tree (RAxML method, 2.7) and cgSNP phylogeny tree (Snippy method, 2.8). In addition to a table of pairwise SNP distance, the authors may add a SNP phylogeny tree for a close comparison.
  • Discussion is too much, needs significant work to focus on the topic. Also, little discussion should be included in conclusion. By the way, why is there a subtitle 4.1, without any other subtitles?

Minor concerns:

  • Abbreviation inconsistency: pneumoniae NDM in some places, and Kp NDM (or KpNDM) in others; some abbreviations never used, e.g. HA, CA in Lines 402-403.
  • Accession dates are required when public genome data were downloaded, because of changes in versions and/or numbers.
  • 6 subtitle could be changed to “whole genome sequencing (WGS) and genome assembling”; while 2.7 subtitle should be changed to “phylogenetic analysis of K. pneumoniae genomes”.
  • Table 1 should be cited in the paragraph of Results 3.1.
  • Line 253: “the strain from 2016 (ND32) and the first strain from 2018 (ND24)”.

Reviewer 2 Report

The authors correlate the metagenomic characteristics of eight strains to confirm the presence of local outbreak and strains unrelated to the epidemic situation. The strains, well characterized,  were isolated from patients with kidney diseases and RTx recipients but it is not clear if  the choice of the isolates studied  is determined by the small number of isolation or by same other specific consideration. Please clarify this question and complete the bibliography.
